🔓 | **Open Peer Review** | Bacteriophages | Research Article

# Profiling the interplay and coevolution of *Microcystis aeruginosa* and cyanosiphophage Mic1

Xiao-Qian Wang,[1,2] Kang Du,[1,2] Chaoyi Chen,[1,2] Pu Hou,[1,2] Wei-Fang Li,[1,2] Yuxing Chen,[1,2] Qiong Li,[1,2] Cong-Zhao Zhou[1,2]

**ABSTRACT**   The cyanosiphophage Mic1 specifically infects the bloom-forming *Microcystis aeruginosa* FACHB 1339 from Lake Chaohu, China. Previous genomic analysis showed that its 92,627 bp double-stranded DNA genome consists of 98 putative open reading frames, 63% of which are of unknown function. Here, we investigated the transcriptome dynamics of Mic1 and its host using RNA sequencing. In the early, middle, and late phases of the 10 h lytic cycle, the Mic1 genes are sequentially expressed and could be further temporally grouped into two distinct clusters in each phase. Notably, six early genes, including *gp49* that encodes a TnpB-like transposase, immediately reach the highest transcriptional level in half an hour, representing a pioneer cluster that rapidly regulates and redirects host metabolism toward the phage. An in-depth analysis of the host transcriptomic profile in response to Mic1 infection revealed significant upregulation of a polyketide synthase pathway and a type III-B CRISPR system, accompanied by moderate downregulation of the photosynthesis and key metabolism pathways. The constant increase of phage transcripts and relatively low replacement rate over the host transcripts indicated that Mic1 utilizes a unique strategy to gradually take over a small portion of host metabolism pathways after infection. In addition, genomic analysis of a less-infective Mic1 and a Mic1-resistant host strain further confirmed their dynamic interplay and coevolution via the frequent horizontal gene transfer. These findings provide insights into the mutual benefit and symbiosis of the highly polymorphic cyanobacteria *M. aeruginosa* and cyanophages.

**IMPORTANCE**   The highly polymorphic *Microcystis aeruginosa* is one of the predominant bloom-forming cyanobacteria in eutrophic freshwater bodies and is infected by diverse and abundant cyanophages. The presence of a large number of defense systems in *M. aeruginosa* genome suggests a dynamic interplay and coevolution with the cyanophages. In this study, we investigated the temporal gene expression pattern of Mic1 after infection and the corresponding transcriptional responses of its host. Moreover, the identification of a less-infective Mic1 and a Mic1-resistant host strain provided the evolved genes in the phage-host coevolution during the multiple-generation cultivation in the laboratory. Our findings enrich the knowledge on the interplay and coevolution of *M. aeruginosa* and its cyanophages and lay the foundation for the future application of cyanophage as a potential eco-friendly and bio-safe agent in controlling the succession of harmful cyanobacterial blooms.

**KEYWORDS**   *Microcystis aeruginosa*, cyanosiphophage, transcriptome, interplay, coevolution

T he accelerated industrialization and urbanization lead to the elevation of $CO_2$ level, global warming, and eutrophication of the natural waterbodies, which cause the uncontrollable growth of cyanobacteria and the seasonal outbreak of cyanobacterial bloom (1–3). The dense blooms bring serious environmental and economic issues,

Address correspondence to Cong-Zhao Zhou, zcz@ustc.edu.cn, or Qiong Li, liqiong@ustc.edu.cn.

The authors declare no conflict of interest.

See the funding table on p. 12.

including the deterioration of water quality, imbalance of ecosystem, and limitations on tourism and fisheries (4–6). Thanks to the capacity of producing gas vesicles, the highly polymorphic *Microcystis aeruginosa* could easily float to an appropriate depth in waterbodies, obtaining enough light and nutrients to become the predominant bloom-forming cyanobacteria (7, 8). Moreover, the lysis of *M. aeruginosa* cells results in the release of microcystin (9), which toxifies the drinking water and threatens the health of surrounding humans and animals (10).

The *M. aeruginosa* blooms have developed in many large freshwater lakes, such as Lake Erie in the United States (11) and Lake Taihu and Chaohu in China (12). These bloom-forming cyanobacteria also harbor diverse and abundant cyanophages, which are a group of bacteriophages that specifically infect and lyse cyanobacteria (13). Actually, abundant CRISPR spacers have been found in the genomes of *M. aeruginosa* (14, 15), indicating the frequent infection of diverse cyanophages. The lytic infection makes the cyanophage a potential eco-friendly and bio-safe agent to control the succession of cyanobacterial blooms (16, 17). *M. aeruginosa* was reported to possess the highest number of antiphage defense genes among the bacteria and archaea (18), suggesting a dynamic interplay and coevolution with its cyanophages in history. However, due to a high-level polymorphism of *M. aeruginosa* (19) and the lack of applicable genetic tools, the detailed processes and mechanisms remain elusive.

Recently we isolated a cyanosiphophage Mic1 from Lake Chaohu, China, which infects *M. aeruginosa* FACHB 1339 (termed *Microcystis* for short). It has an icosahedral capsid of ~88 nm in diameter, followed by a long flexible tail of ~400 nm (20). Whole-genome sequencing showed that Mic1 possesses a double-stranded DNA (dsDNA) genome of 92,627 bp, containing 98 putative open reading frames (ORFs) (21). Using transcriptome sequencing, here, we profiled the temporal gene expression pattern of Mic1 during infection and analyzed the transcriptional responses of *Microcystis*. Moreover, the identification of a less-infective Mic1 and a Mic1-resistant host strain enabled us to explore their coevolution in the laboratory. These findings help us better understand the interplay between *M. aeruginosa* and its cyanophages and lay the foundation for the future application of cyanophages in controlling the harmful blooms.

## RESULTS

### The transcriptome dynamics of Mic1 and *Microcystis* through infection

We first measured the one-step growth curve by applying Mic1 to infect *Microcystis* cells at a multiplicity of infection (MOI) of 3. The results showed that nearly all host cells were rapidly recognized by Mic1 particles within 2 h of infection, and no multiple-infection phenomenon occurred (Fig. 1A). From 6–8 h post infection, the progeny Mic1 began to release from the infected host cells to the culture medium; and after 8 h of infection, a fast-increasing number of extracellular Mic1 was detected (Fig. 1A). In addition, along with the release of progeny Mic1 and the lysis of infected cells, the number of host cells tend to decrease after 10 h of infection (Fig. 1A).

Accordingly, total RNA samples at various time points after infection were extracted and applied to transcriptome sequencing, respectively. After data cleaning, the mRNA reads were mapped to the genomes of Mic1 and *Microcystis*, respectively. The transcriptome dynamics revealed a slow accumulation of Mic1 transcripts, starting from ~0.5% at 2 h and reaching the peak of ~19.2% at 10 h post infection (Fig. 1B). It indicated that the transcription of Mic1 genes starts within 2 h, and massive progeny phages are released after 10 h of infection. Meanwhile, *Microcystis* transcripts gradually decreased along with the infection time, but the remaining accounted for 80.8% of total transcripts at 10 h post infection (Fig. 1B). Moreover, the principal component analysis showed that the transcripts of Mic1 at eight time points can be classified into three groups (Fig. S1A), corresponding to the various phases in the phage lytic cycle. Compared to the uninfected *Microcystis*, the infected cells display a quite different transcriptional profile, and the difference is gradually enlarged along the infection process (Fig. S1B). Notably,

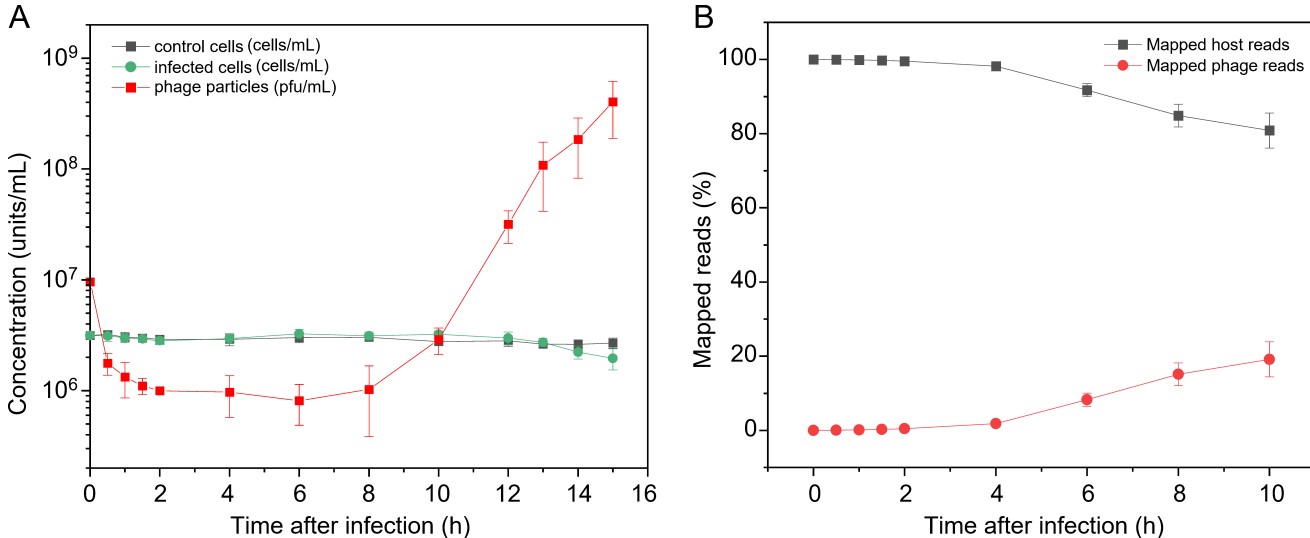

**FIG 1** The transcriptome dynamics of Mic1 and *Microcystis* through infection. (A) The one-step growth curve of Mic1 infecting the host *Microcystis* at a MOI of 3. The control cells are treated with an equal volume of BG11 medium instead of Mic1 lysate. (B) Ratios of Mic1 and *Microcystis* transcripts at different time points post infection were determined from RNA-seq reads that mapped to the phage and host genomes, respectively. Data are presented as mean ± SD from three independent experiments.

all these transcriptome dynamics results of Mic1 and *Microcystis* are consistent with the one-step growth curve of the infection process.

## Clustering and temporal expression pattern of Mic1 genes

Similar to the previous reports of dsDNA phages (22, 23), statistic analysis against the abundance of Mic1 transcripts showed that 98 putative genes of Mic1 could be preliminarily clustered into three temporal phases (Fig. 2A): early (0.5–2 h), middle (2–6 hr), and late (6–10 h). The early phase contains 23 genes from *gp49* to *gp73* (excluding *gp61* and *gp67*) in the same transcription direction (Fig. S2), which might play a role in the take-over of host transcription and metabolism. The middle phase consists of 31 genes involved in DNA replication and nucleotide metabolism, whereas the late phase comprises 44 genes necessary for the assembly of progeny phages and the lysis of host cells (Fig. S2).

Mfuzz analysis (24) revealed that the genes in each phase could be further grouped into two clusters (termed E1&E2, M1&M2, and L1&L2, respectively), which display distinct temporal expression patterns (Fig. 2B; Table S1). Six early genes, *gp49* and *gp69–gp73*, constitute the E1 cluster, the transcriptional level of which immediately reached the peak at 0.5 h and then quickly decreased to the lowest at 4 h post infection (Fig. 2B). Searching against the PHROGs database (25), the proteins encoded by *gp69* and *gp70* are annotated as secreted proteins closely related to the superinfection exclusion. The proteins gp71 and gp73 are homologous to a lysine tRNA synthetase C-terminal domain and helix-turn-helix domain, respectively, which play a potential role in binding to RNA and DNA (26, 27). The *gp72* gene encodes a hypothetical protein, which adopts a novel fold in the α + β class (28). The immediate expression of these E1 genes might be favored for Mic1 to rapidly regulate and redirect the host transcription and metabolism toward the amplification of progeny phages.

Besides the six E1 genes, the remaining 17 early genes belong to the E2 cluster (Table S1). In general, the genes in this cluster are highly expressed at 1 h post infection and remain at a high transcriptional level until 2 h post infection, followed by a quick decline along the infection time (Fig. 2B). The E2 cluster includes genes coding for the transposase (*gp52*), DNA-binding protein (*gp59*), and Rho termination factor N-terminal domain-containing protein (*gp63*), in addition to several hypothetical proteins of

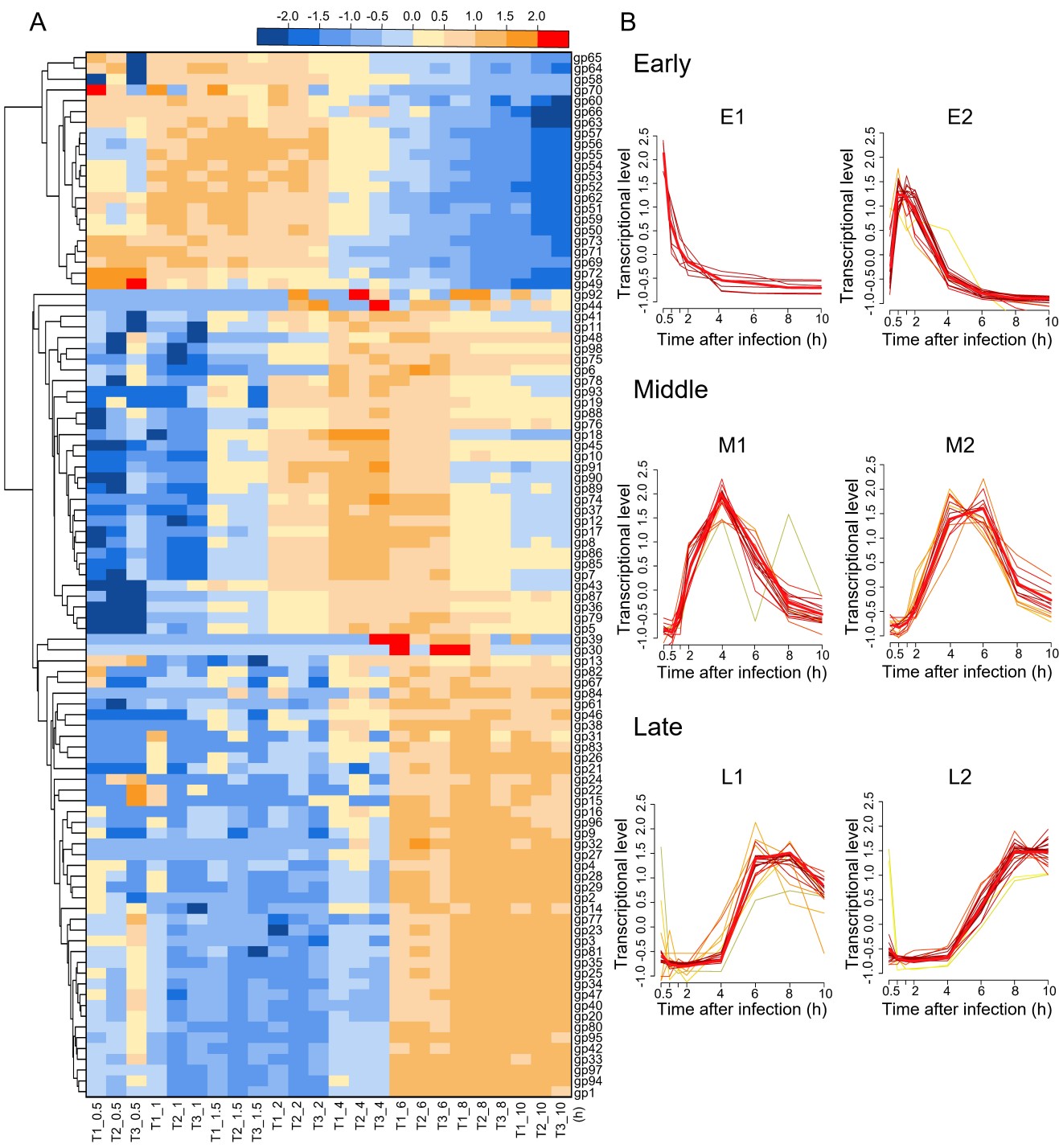

**FIG 2** Clustering and temporal expression pattern of Mic1 genes. (A) Hierarchically clustered heat map showing the clustering of Mic1 genes into early, middle, and late temporal phases. Based on the $Log_2(TPM + 1)$, hierarchical clustering was performed using Euclidean distance and average linkage metrics as implemented in the Heat Map Dendrogram of OriginPro 2020. Different colors represent distinct transcriptional levels of Mic1 genes, which change along the time course. Three independent infection samples are indicated with T1, T2, and T3, whereas the number following the "_" represents the time point post infection. (B) Each temporal phase of Mic1 was further grouped into two clusters by R package Mfuzz, based on the transcriptional levels of various genes. The genes in each cluster share a similar temporal expression pattern, with the red line highlighting the common transcription tendency. The number of reads for each gene at a specific time point is the average of three independent experiments. The individual genes in each cluster are listed in Table S1.

unknown function (Table S1). These proteins might further enable Mic1 to hijack the transcription and metabolism of the host.

The transcriptional levels of 18 genes in the M1 cluster displayed as a sharp peak with the highest level at 4 h post infection, whereas those of 14 genes in the M2 cluster remained highly expressed from 4 to 6 h post infection (Fig. 2B; Table S1). In the M1 cluster, gp10/DNA primase and gp85-gp86/DNA polymerase gamma are required for DNA replication, whereas gp12/RNA ligase, gp89/DNA helicase, and gp92/HNH endonuclease might participate in the nucleotide recombination and repair (Table S1). Besides, the M1 cluster contains two genes, *gp18* and *gp19*, encoding chaperones, the transcription of which is preparing for the tail assembly of progeny phages. The M2 cluster also comprises genes coding for the flap endonuclease (*gp6*), RNA polymerase sigma-70 factor (*gp36*), and DNA endonuclease I-CreI (*gp87*) in the DNA replication and repair, in addition to the thymidylate synthase (*gp37*) and ribonucleotide reductase (*gp75*) in the nucleotide metabolism (Table S1). Moreover, the expression of a host-derived auxiliary metabolic gene (AMG) *mazG* (*gp98*) might allow the hydrolysis and recycling of the host DNA, facilitating the replication of the phage genome (29).

The late genes constitute 44% of Mic1 ORFs, all of which possess a drastically increased transcriptional level after 4 h of infection (Fig. 2B). In detail, 19 genes in the L1 cluster reached the highest transcriptional levels at 6 h, which were then declined at 8 h post infection (Fig. 2B). The proteins encoded by these genes include gp9/N-acetyl-muramidase, gp30/peptidoglycan endopeptidase, and gp80/L,D-transpeptidase (Table S1), which might be responsible for the lysis of host. The transcription of genes *gp1*, *gp16*, *gp31*, and *gp32*, coding for a terminase large subunit, receptor-binding protein, phage tail L, and min tail, respectively (Table S1), initiates the DNA packaging and tail assembly of progeny Mic1. The DNA adenine methylase (gp14) expressed in this cluster, together with DNA (cytosine-5)-methyltransferase (gp90) expressed in the M1 cluster (Table S1), might both help methylate the nascent Mic1 genome, preventing the degradation by the host restriction endonucleases in the next cycle of infection (30). In addition, the expression of another AMG *phoH* (*gp97*) in the L1 cluster might alter host metabolism and provide energy for the assembly of progeny phages, as PhoH is involved in regulating phosphate uptake (31). Therefore, the transcriptional levels of genes that are involved in the assembly increased from 4 to 8 h and remained high until the host lysis at 10 h post infection (Fig. 2B). The L2 cluster contains 11 genes encoding the structural proteins of Mic1, in addition to *gp13* (ATP-binding protein), *gp81* (clp protease), and 11 function-unknown genes (Table S1). Notably, the expression of a head scaffolding protein (gp42) in this cluster suggested that the tail assembly for the long-tailed phages might be initiated independently in prior to the assembly of the capsid. Altogether, upon infection, the temporal expression of Mic1 genes in the host cells ensures its amplification and fulfillment of the whole lytic cycle.

## Transcriptomic responses of *Microcystis* upon Mic1 infection

Compared to the control cells, the growth of infected *Microcystis* cells is not obviously altered in 10 h after Mic1 infection (Fig. 1A). However, during this period, various host transcriptomic responses should have been triggered. Genome sequencing showed that *Microcystis* possesses a 4.7 Mb dsDNA genome containing 4,519 ORFs. Upon the infection of Mic1, the number of differentially expressed genes (DEGs) increased, up to ~24% of *Microcystis* ORFs at 6 h post infection (Table S2; Fig. S3). Notably, the transcriptional levels of host genes in the infected cells were compared with those in the control cells at each time point in order to distinguish the phage-induced changes from the growth-dependent changes of the host itself.

In-depth analysis of the DEGs revealed that genes in a polyketide synthase pathway were significantly upregulated from 0.5 to 1.5 h post infection (Fig. 3A; Table S3), of which a hypothetical regulator ORF2639 might be responsible for the transcriptional regulation of this pathway. The immediate upregulation of these genes might enable the host to synthesize various metabolites for the rapid defense against the infection of

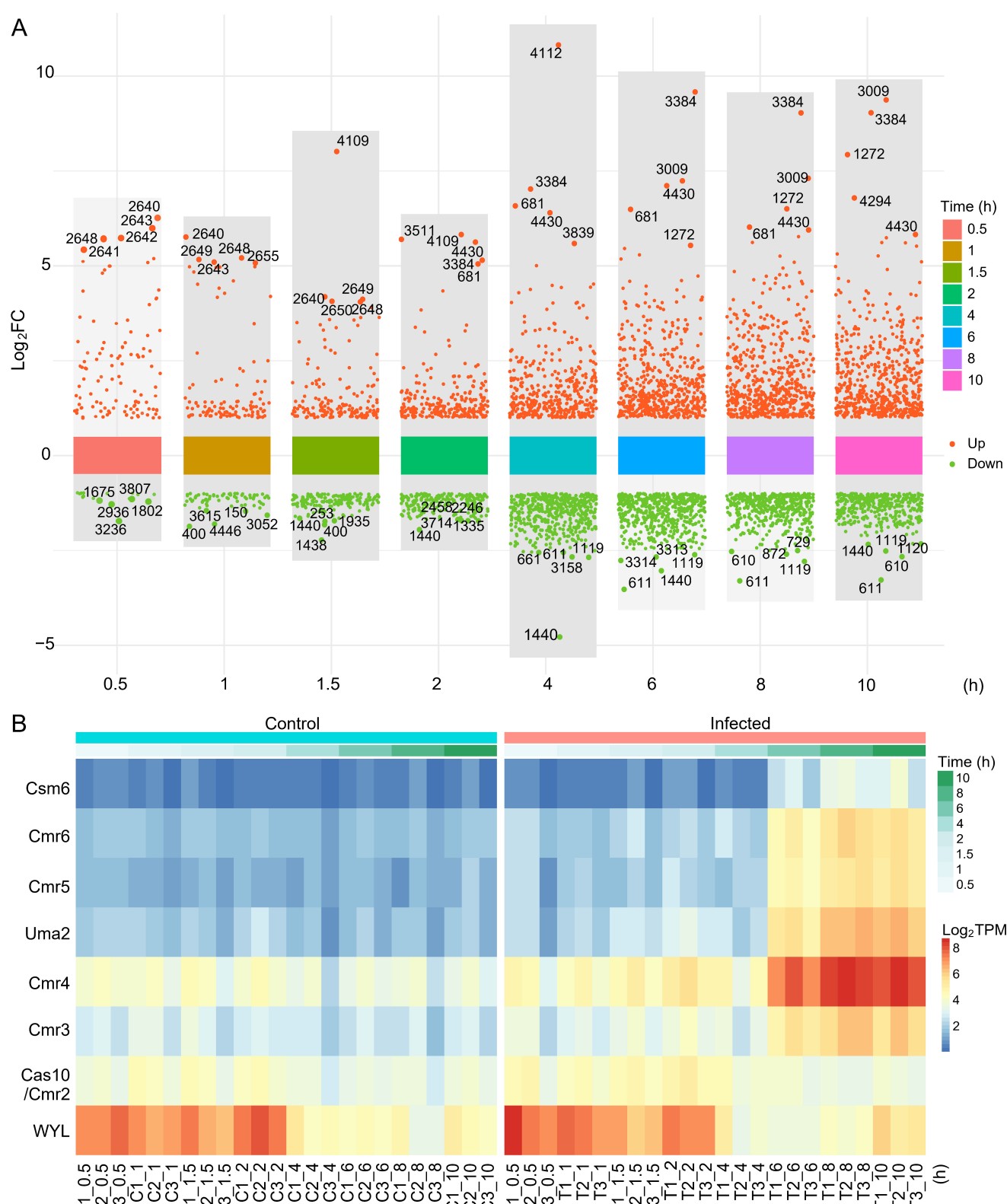

**FIG 3** Transcriptomic responses of host *Microcystis* upon Mic1 infection. (A) An overall visualization of host DEGs at different time points. The y-axis represents the Log$_2$FC for each DEG, whereas the x-axis indicates the time points after infection. The red and green dots represent genes with upregulated and downregulated transcriptional levels, respectively. The top five upregulated and downregulated DEGs at each time point are labeled. (B) Heat maps showing the

**FIG 3** (Continued)

transcriptional levels of DEGs encoding type III-B CRISPR system in infected (T) and control (C) host cells. Different colors represent distinct transcriptional levels, which change along the time course. The graph is drawn by ImageGP (https://www.bic.ac.cn/ImageGP/index.php/Home/Index/PHeatmap.html), based on the $\text{Log}_2\text{TPM}$.

Mic1. Moreover, most genes of a type III-B CRISPR system showed obviously upregulated transcriptional levels after 6 h of infection (Fig. 3B; Table S4), likely representing a host defense against the potential lysogenic cycle of Mic1 that was identified in a previous report (21). In addition, the upregulation of *orf4430* (phycobilisome degradation protein NblA) and *orf681* (phycobiliprotein lyase) from 2 to 10 h post infection (Fig. 3A) would initiate the degradation of the cyanobacterial light-harvesting complex phycobilisome, which is an adapted response to nutrient limitation (32), meanwhile providing amino acids for the synthesis of phage proteins (33). The transcription of the methylase DpnIIA (*orf3009*) was upregulated at 6–10 h post infection (Fig. 3A), which might be embezzled by Mic1 to augment the methylation of its nascent genomic DNA.

Compared to the upregulated genes, the transcriptional downregulation of host genes seemed to occur later, most of which possess a relatively lower fold-change value (Fig. 3A). In fact, at 0.5 h post infection, only 17 genes were downregulated with a $\text{Log}_2$ Fold Change ($\text{Log}_2\text{FC}$) between −1 and −1.8, in contrast to that totally 109 genes were upregulated, 57 of which have a $\text{Log}_2\text{FC} \geq 1.8$ with a *P*-adj $\leq 0.05$ (Fig. 3A). It suggested that Mic1 prefers to take over and utilize the host resource to fulfill its amplification and not simply shut down the host metabolism. KEGG pathway enrichment analysis against all the downregulated genes revealed that the most enriched pathway is usually associated with photosynthesis, whose downregulation occurred at 1.5 h post infection and remained at a low level until the lysis of host cells (Table S5). Meanwhile, the downregulated genes encoding proteins of the porphyrin and chlorophyll metabolism were enriched after 2 h of infection (Table S5). Notably, genes involved in carbohydrate, lipid, and amino acid metabolism, in addition to oxidative phosphorylation and carbon fixation, were also downregulated upon the infection of Mic1 (Table S5). The moderate downregulation of these pathways would enable the host to survive for providing resources for the production of massive progeny Mic1 particles via Mic1-hijacked metabolism pathways.

## Horizontal gene transfer and coevolution of Mic1 and host *Microcystis*

In fact, the crosstalk between the host *Microcystis* and its cyanophage Mic1 remains active during the multiple-cycle infection and amplification in the laboratory. We found a Mic1-resistant strain on the solid medium and re-sequenced its genome. Compared to the genome of the original *Microcystis*, the genome of Mic1-resistant strain possesses two single nucleotide variants, one insertion-deletion frameshift and one copy number variation (CNV) (Table 1). In detail, base substitutions G323T of *orf418* and C528G of *orf3855* cause the nonsynonymous mutations of G108V in gas vesicle protein K and H176Q in bifunctional (p)ppGpp synthetase-hydrolase, respectively. A frameshift deletion terminates the expression of ORF2938, which is structurally similar to a conserved biofilm-related protein Se0862 from *Synechococcus elongatus* PCC 7942 (Fig.

**TABLE 1** Sequence variations in the genome of Mic1-resistant strain

| Mutation type | Gene | Position | Gene | Protein | Annotation |
|---|---|---|---|---|---|
| Nonsynonymous SNV[a] | *orf418* | 453,985 | G323T | G108V | Gas vesicle protein K |
| Nonsynonymous SNV[a] | *orf3855* | 4,079,165 | C528G | H176Q | Bifunctional (p)ppGpp synthetase/guanosine-3',5'-bis(diphosphate) 3'-pyrophosphohydrolase |
| Frameshift deletion | *orf2938* | 3,102,962 | 267_270del | I89fs | Biofilm-related protein |
| CNV duplication | *orf3030–orf3050* | 3,199,701–3,214,800 | / | / | / |

[a]SNV: Single nucleotide variation.

S4). In addition, a 15,100 bp duplication of *orf3030–orf3050*, including genes that encode transposase, integrase, and peptidase (Table S6), was found between the 3′-terminal of *orf3029* with an overlap of 45 bp and the 5′-terminal of *orf3051* with an overlap of 57 bp. Notably, as shown in our transcriptome sequencing data, the transcription levels of three genes *orf418*, *orf3855*, and *orf2938* remained unchanged, whereas those of several genes in *orf3030–orf3050* were significantly increased (Table S6). These changes in the *Microcystis* genome might enable the strain to resist the infection of Mic1.

In addition, we also isolated a less-infective strain of Mic1 toward the *Microcystis* host, which possesses a much lower burst size (~2 PFU/cell) compared to that (~450 PFU/cell) of the original Mic1. Resequencing the genome of this Mic1 strain revealed a couple of mutations, most of which are neutral base substitutions except for an insertion in the coding region of *gp49* gene. AlphaFold2 prediction (34) combined with DALI search (35) revealed that gp49 of Mic1 is structurally similar to TnpB (Fig. S5A), with a root-mean-square deviation (RMSD) of 2.8 Å over 305 Cα atoms. As recently reported (36), TnpB is an RNA-guided DNA endonuclease and the predecessor of type V CRISPR endonuclease Cas12. Moreover, the multiple-sequence alignment analysis showed that gp49 has a primary sequence similarity of 69% with TnpB, sharing the highly conserved catalytic residues (Fig. S5B). It suggested that gp49 is a TnpB-like transposase and might also function as an endonuclease. The E1 gene *gp49* of the less-infective Mic1 was inserted with a full-length host gene *orf2067*, in addition to its upstream 207 bp and downstream 59 bp bases (Fig. S5C). Notably, via structure prediction and comparison, we found that the host ORF2067 shares a similar structure to the nickase IsrB (Fig. S6), with an RMSD of 2.3 Å over 315 Cα atoms. This insertion led to the loss-of-function truncation of gp49, which lacks the key DNA-binding structural element. It represents a host-driven beneficial evolutionary event due to the frequent interplay and horizontal gene transfer during the infection and defense between Mic1 and *Microcystis*.

## DISCUSSION

Upon infection, the phage tends to shut down the transcription of some host genes and hijack the necessary host metabolism pathways for the amplification of progeny phages (37). In response to the infection, the host activates the defense systems and/or gains the capability of phage resistance (38, 39). To date, most investigations have been focused on marine cyanobacteria and corresponding cyanophages (40), except for a recent report on the freshwater cyanobacteria *M. aeruginosa* that revealed the expression pattern of cyanomyophage Ma-LMM01 and the host responses (41).

Here, we performed transcriptome sequencing of *Microcystis* upon the infection of cyanosiphophage Mic1 and revealed the phage lytic cycle and the corresponding host transcriptional responses. The 98 ORFs of Mic1 could be grouped into six distinct clusters (Fig. 2). Similar to the annotations of Ma-LMM01 and marine *Synechococcus* phage Syn9 (41, 42), Mic1 also encodes two putative kinases (gp45 and gp74) and a hypothetical halogenase (gp91) in the middle phase. However, genes coding for the terminase and lysozyme in Ma-LMM01 were classified into the middle genes (41), whereas those are late genes for Mic1.

Moreover, a comparison of the global transcriptomic profiles of the phage and host revealed two distinct infection modes according to the increased velocity of phage transcripts: constant and sudden modes (Fig. 4). Similar to *M. aeruginosa* cyanophage Ma-LMM01 (41), Mic1 also employs a constant increase mode post infection. The transcripts of Mic1 only reach a final replacement rate of ~20%, and Ma-LMM01 possesses a relatively low replacement rate of ~33% as well (41). The very low replacement rate of Mic1 might be necessary to balance the responses of hundreds of host DEGs, as only eight host DEGs were activated by Ma-LMM01 (41). In contrast, the marine *Prochlorococcus* cyanophage P-HM2 and *Synechococcus* cyanophage Syn9 both adopt a sudden increase mode, with a transcript replacement rate of 65% and 98%, respectively (42, 43). Notably, bacteriophages, such as PAK_P3 (44) and PAK_P4 (45) infecting *Pseudomonas aeruginosa*, also use these two distinct infection modes, respectively,

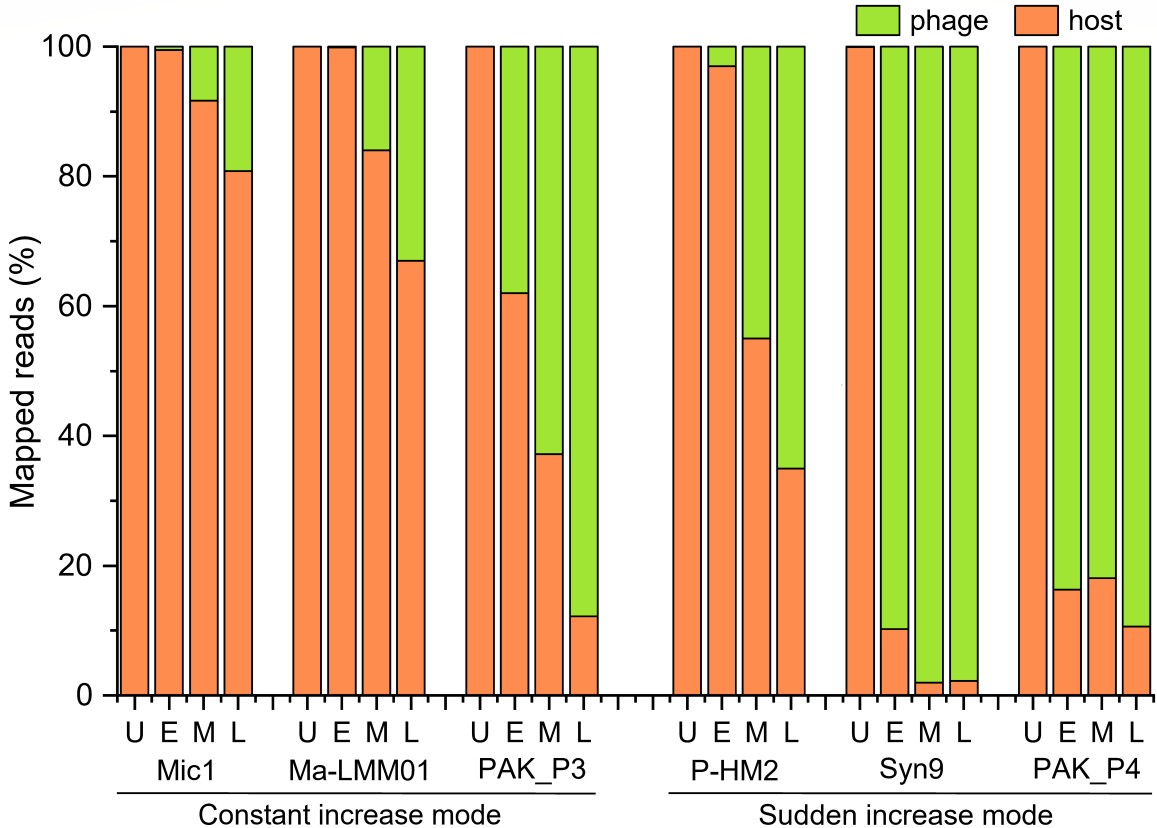

**FIG 4** Alignments of RNA-seq reads sets against host or phage genome at different infection phases. The green bar represents the percentage of transcripts that map to the phage genome, whereas the orange bar represents the percentage of transcripts that map to the host genome. Besides Mic1, the reads percentages for other cyanophages/phages are shown for comparison, including Ma-LMM01 (41), Syn9 (42) , P-HM2 (43), PAK_P3 (44), and PAK_P4 (45), whose reads are collected from previously published data. U: uninfected; E: early phase; M: middle phase; L: late phase.

despite they both display a high replacement rate of ~90%. In addition, the transcription of a lysogenic-related ParABS plasmid partition system further supports the existence of a lysogenic cycle for Mic1. All these indicate that Mic1 might utilize a unique strategy, constant increase combined with a low replacement rate and a potential lysogenic cycle, to mildly control the host genome for longer and better co-existence and coevolution with the highly polymorphic host *M. aeruginosa*.

Multiple populations of cyanophages and hosts permanently coexist in the blooms, inevitably leading to the extensive exchange of genetic materials through the infection and defense responses. A loss-of-function horizontal gene transfer of host *orf2067* encoding a homolog of IsrB nickase into Mic1 *gp49* encoding a TnpB-like transposase causes the decrease of Mic1 infectivity. Previous studies revealed that IsrB, a member of the IS200/IS605 superfamily of transposons, is usually responsible for nicking the non-target strand of dsDNA containing a 5′-NTGA-3′ target-adjacent motif (46). Indeed, *orf2067* is inserted at the motif 5′-ATGA-3′, corresponding to 263–266 of original *tnpB*. On the other hand, a couple of mutations in the genome enable the *Microcystis* to gain resistance against Mic1, probably via decoration of host receptors and/or activation of host defense systems. In fact, a previous report of *S. elongatus* PCC 7942 proposed that the frameshift inactivation of *Microcystis* ORF2938 might contribute to the formation of biofilm (47), which likely impedes the recognition of Mic1 against the receptors at the host cell surface.

## MATERIALS AND METHODS

### *Microcystis* and Mic1 culture conditions

The host *Microcystis*, bought from the Freshwater Algae Culture Collection at the Institute of Hydrobiology (Wuhan, China), was grown in BG11 medium at 28°C under a light intensity of 2,000 lux with 14/10 h light/dark photoperiod to an $OD_{680\ nm}$ of 0.8.

To prepare the phage lysate, 100 mL *Microcystis* cells at the exponential growth phase were infected with Mic1 at an MOI of 1 and then cultured in the same condition for another 2 days. The harvested lysate was filtered through 0.2 µm polycarbonate membrane filters and stored as a seed lysate at 4°C.

For the transcriptome sequencing, *Microcystis* cells were cultured in a 400 mL BG11 medium to an $OD_{680\ nm}$ of 0.8 as described above. The cell number was counted by flow cytometry (BD) based on scatter and algae's autofluorescence according to the protocol by Marie et al. (48) and adjusted to an initial density of $3 \times 10^6$ cells/mL for infection. The Mic1 titer was determined by plaque assays on the solid medium as previously reported (20). To obtain simultaneously infected cells without multiple infections, cultured *Microcystis* cells were infected with Mic1 seed lysate at an MOI of 3. For the control cells, an equivalent volume of BG11 medium was added instead of the Mic1 lysate. Three independent infection experiments were performed.

### Sequencing and analysis of *Microcystis* genome

The host *Microcystis* was cultured to an $OD_{680\ nm}$ of 0.8 as mentioned above, and 2 g fresh cells were collected by centrifugation. Subsequently, the genomic DNA was extracted and sequenced by the next-generation sequencing strategy under the Illumina MiSeq platform in combination with the third-generation single molecule sequencing strategy under the PacBio Platform (Shanghai Personal Biotechnology Co., Ltd., China). After removing the adapters and poor-quality reads, all the clean reads were assembled by HGAP4 (49) and Canu softwares (50), followed by correction via Pilon software (51). The *de novo* assembled *Microcystis* genome was further annotated using Prokka software (52) and then confirmed by BLASTp (https://blast.ncbi.nlm.nih.gov/Blast.cgi) against the National Center for Biotechnology Information and HHpred (https://toolkit.tue-bingen.mpg.de/tools/hhpred), respectively. The putative CRISPR in the genome was identified by CRISPRFinder (53).

### RNA-seq experimental design and sample collection

A one-step growth curve for Mic1 under an MOI of 3 was performed according to our previous study (20), except that the cell number of *Microcystis* was determined by flow cytometry. The growth curve was drawn by the OriginPro 2020 software, and three biological replicates were used to calculate the means and SDs.

Then, based on the growth curve, 30 mL samples were collected from the infected and control cells at nine time points (0, 0.5, 1, 1.5, 2, 4, 6, 8, and 10 h after the addition of phage/BG11), respectively, via centrifugation at 10,000 g for 10 min at 4°C. The cell pellets were washed twice with 1 mL ice-cold Phosphate buffered saline, quickly frozen in liquid nitrogen, and stored at −80°C for the subsequent RNA extraction. For each time point, three replicate samples were respectively collected from three infection experiments.

### RNA extraction, sequencing, and data processing

Total RNA was extracted from the frozen cell pellets using the Promega Total RNA Isolation kit (Eastep REF ls1040) following the manufacturer's instructions. The Qubit Fluorometer (Thermo Fisher Scientific, USA) was used to measure the concentration of extracted total RNA, and the RNA integrity was analyzed by the Agilent 2100 Bioanalyzer (Agilent Technologies, USA). After removing rRNA using the Ribo-off rRNA Depletion Kit (Bacteria; Vazyme #N407, China), the enriched mRNA was fragmented by

high temperature. Subsequently, reverse transcription and second-strand synthesis were sequentially performed, obtaining dsDNA. Via T4 DNA ligase, the dTTP-tailed adaptor was ligated to both ends of the dsDNA fragments, which were then amplified by PCR and circularized to obtain a single-stranded circular (ssCir) library. After quality control, the ssCir library was then amplified through rolling circle amplification to obtain more than 300 copies of DNA nanoball, which was further loaded onto the patterned array chip and sequenced by the DNBSEQ platform at the China National GeneBank.

FastQC (54) was employed to evaluate the quality of raw-sequencing data, which was then trimmed by Trim galore ([https://www.bioinformatics.babraham.ac.uk/projects/trim_galore/](https://www.bioinformatics.babraham.ac.uk/projects/trim_galore/)) to remove bar codes and adapters. Then, host rRNA reads were removed manually by bowtie2 (55) prior to read mapping. Using STAR-RSEM (56), the clean reads were aligned separately to the viral and host reference genomes, counted for each gene, and normalized as TPM (transcripts per kilobase of exon model per million mapped reads).

## Clustering of phage gene expression

The TPM derived from the RNA-seq data was used for the cluster analysis of Mic1 gene expression patterns. Hierarchical clustering analyses were performed using Euclidean distance and average linkage metrics as implemented in the Heat Map Dendrogram of OriginPro 2020. When plotting the heat map with a dendrogram, "Rows" were selected for standardization and clustering. Moreover, the R package Mfuzz (57) was used to further classify the Mic1 gene expression patterns. Notably, Mfuzz is a method for time trend analysis of transcriptional changes, whose core algorithm is based on Fuzzy C-Means Clustering (24, 57).

## Identification of differentially expressed genes of the host

Transcriptional levels of host genes were analyzed separately at each time point, comparing the infected and control host cells. DEGs were identified using the R package DESeq2 (58) with the default parameters. Significant DEGs were defined with a $P$-adjust value ($P$-adj, $P$-value with a multiple-test correction) < 0.05 and a $Log_2FC$ value ≥ |1|. The raw data were listed in Table S7. Using R package dplyr (59), all the significant DEGs were selected and assigned with the tag of upregulation or downregulation. Notably, the top five upregulated and downregulated genes at each time point were especially picked. Finally, the significant DEGs from different time points were simultaneously displayed in a multigroup plot using R package ggplot2 (60).

## Screening and genome resequencing of the resistant strain

A mixture of *Microcystis* and Mic1 was spread on a solid plate and cultivated in the incubator at 28°C under a light intensity of 2,000 lux with 14/10 h light/dark photoperiod. After forming plaques due to the infection, the solid plate was continuously cultivated in the incubator under the same condition for more than 2 weeks. Once appearing at the plaque, new *Microcystis* clones were selected for culturing in liquid BG11 medium. After several rounds of iterative screening, a resistant strain against the infection of Mic1 was obtained.

The genome of the resistant strain was extracted and applied for resequencing by the next-generation sequencing strategy. TruSeq DNA PCR-free prep kit was used to prepare the sequencing library with inserts of 400 nt in length. The library quality and concentration were assessed using Agilent bioanalyzer and Quant-it PicoGreen dsDNA Assay Kit, respectively. Final libraries were sequenced on Illumina NovaSeq sequencing platform (Shanghai Personal Biotechnology Co., Ltd., China) with a paired-end read length of 2 × 150 bp. After evaluating the sequencing quality by FastQC (54), the resequencing reads were aligned to the genome of wild-type *Microcystis* by bwa (61). SNP and InDel (Insertion and Deletion) were analyzed by GATK (62), whereas CNV was analyzed by CNVnator (63). All the genetic variants were verified by PCR. The three-dimensional

structures of specific proteins were predicted by AlphaFold2 (38). Multiple-sequence alignment was performed using the Multalin program (64).

## ACKNOWLEDGMENTS

We thank Dr. Ziqing Deng at the China National GeneBank (CNGB) and Professor Qinglu Zeng at the Hong Kong University of Science and Technology for their technical support on the RNA-seq data collection and analysis.

This research was supported by the National Natural Science Foundation of China (grant nos. 32000112 and U19A2020), the Ministry of Science and Technology of China (grant no. 2018YFA0903100), and the Fundamental Research Funds for the Central Universities (WK2070000195).

C.-Z.Z. and Q.L. conceived, designed, and supervised the project. X.-Q.W., Q.L., and C.C. analyzed the data. Q.L., C.-Z.Z., and X.-Q.W. wrote and revised the manuscript. X.-Q.W. and K.D. constructed the data processing and analysis platform. X.-Q.W., P.H., and Q.L. performed the sample collection and extraction. W.-F.L. and Y.C. offered excellent suggestions on the data analysis. All of the authors discussed the data and read the manuscript.

## AUTHOR AFFILIATIONS

[1]School of Life Sciences, Division of Life Sciences and Medicine, University of Science and Technology of China, Hefei, China
[2]Biomedical Sciences and Health Laboratory of Anhui Province, University of Science and Technology of China, Hefei, China

## AUTHOR ORCIDs

Qiong Li http://orcid.org/0000-0003-2838-9380
Cong-Zhao Zhou http://orcid.org/0000-0002-6881-7151

## FUNDING

| Funder | Grant(s) | Author(s) |
| --- | --- | --- |
| MOST \| National Natural Science Foundation of China (NSFC) | 32000112 | Qiong Li |
| MOST \| National Natural Science Foundation of China (NSFC) | U19A2020 | Cong-Zhao Zhou |
| Ministry of Science and Technology of the People's Republic of China (MOST) | 2018YFA0903100 | Cong-Zhao Zhou |
| MOE \| Fundamental Research Funds for the Central Universities (Fundamental Research Fund for the Central Universities) | WK2070000195 | Qiong Li |

## DATA AVAILABILITY

The raw RNA sequencing data of uninfected/infected Microcystis aeruginosa FACHB 1339 at nine time points have been deposited in the Sequence Read Archive database under the accession numbers of SRR27490991–SRR27491041. The raw genome resequencing data of the resistant host strain has been deposited in the Sequence Read Archive database under the accession number of SRR27490419. The genome of original Microcystis aeruginosa FACHB 1339 has been deposited in the GenBank database with the accession number of CP142375. All data generated or analyzed during this study are included in the manuscript and its supplementary information files.

## ADDITIONAL FILES

The following material is available online.

### Supplemental Material

**Supplemental figures and tables (Spectrum00298-24-s0001.pdf).** Fig. S1-S6; Tables S2-S6.
**Table S1 (Spectrum00298-24-s0002.xlsx).** Clustering and temporal expression pattern of Mic1 genes.
**Table S7 (Spectrum00298-24-s0003.xlsx).** Transcriptional level of host Microcystis genes after Mic1 infection.

### Open Peer Review

**PEER REVIEW HISTORY (review-history.pdf).** An accounting of the reviewer comments and feedback.

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
