## [Reviewer comments · Microbiology Spectrum]

Microbiology Spectrum

Profiling the interplay and coevolution of *Microcystis aeruginosa* and cyanosiphophage Mic1

Xiao-Qian Wang, Kang Du, Chaoyi Chen, Pu Hou, Wei-Fang Li, Yuxing Chen, Qiong Li, and Cong-Zhao Zhou

Corresponding Author(s): Cong-Zhao Zhou, University of Science and Technology of China

Review Timeline:

Submission Date:	January 31, 2024
Editorial Decision:	February 28, 2024
Revision Received:	March 27, 2024
Accepted:	April 5, 2024

Editor: Blaire Steven

Reviewer(s): Disclosure of reviewer identity is with reference to reviewer comments included in decision letter(s). The following individuals involved in review of your submission have agreed to reveal their identity: Konstantin V. Severinov (Reviewer #1)

Transaction Report:

DOI: <https://doi.org/10.1128/spectrum.00298-24>

Re: Spectrum00298-24 (Profiling the interplay and coevolution of *Microcystis aeruginosa* and cyanosiphophage Mic1)

Dear Prof. Cong-Zhao Zhou:

Thank you for the privilege of reviewing your work. Below you will find my comments, instructions from the Spectrum editorial office, and the reviewer comments.

This manuscript was reviewed and seen and worthy of publication after a few minor modifications. Overall, I found the manuscript well written and presented, although I did find some minor errors in English and grammar such as the following:
Line 43: I would remove "in the long history"

Line 47 replace has ever been with "is"

Line 79 replace been burst with "developed"

If possible I would suggest the authors have someone proficient in English to check through, but again I would note the errors are small and do not take away from understanding the science of the manuscript.

Revision Guidelines

Sincerely,

Reviewer #1 (Comments for the Author):

Freshwater cyanobacteria can cause devastating blooms and their interaction with phages is of considerable scientific and practical interest. In the manuscript, Wang et al. investigated the transcriptome of the cyanobacterium *Microcystis aeruginosa* infected with a previously isolated and sequenced siphovirus Mic1. Based on time-course changes in transcript abundances, phage genes were divided into three classes (early, middle and late) with two subclusters within each group. The authors assigned plausible putative functions for many genes from each temporal classes. They also monitored changes in abundance of host transcripts at different stages of viral infection. A Mic1 phage resistant *M. aeruginosa* strain, as well as an alleviated phage were isolated and mutations responsible were identified.

Below a several comments/questions for the authors.

1. The two separate axes as well as separate scales for the host cell number and PFU/ml in Fig. 1a are confusing. Since the logarithmic scale is used, both could be depicted on the same scale.
2. Did the authors analyze the structure of viral DNA within the virions, specifically, were the termini of viral DNA determined? In Fig. S2 the genome map starts with the *tnpB* gene. Does it imply that this gene is truly located at the terminus? Given that *tnpB* is an early gene, does this imply that this end is injected first?
3. It is intriguing that Mic1 genome contains quite large apparently non-coding regions. For instance, in the published annotation ~3 kbps region downstream *tnpB* (gene 49) lacks any annotated features. Given that most phage genomes demonstrate very high coding density, it is quite puzzling. Do the authors have any clues on the functional role of this region? Is this region transcriptionally active?
4. It is stated that it Fig. 3A contains a volcano plot. Usually, a volcano plot shows two values: significance and fold change. In Fig. 3A, only log₂ fold change axis is shown without the significance. Either the plot should be corrected or the text modified.
5. Replace gene numbers with the names of cas genes (e. g. *csm6* instead 1109) in Fig. 3b.
6. In the Materials and Methods section it is stated that differentially expressed host genes were identified using DESeq2 package. Results of the analysis, including adjusted p-values, should be provided. Currently, only log₂FC values (e. g., Table S3) are shown.
7. Line 262 states "we also isolated a strain of Mic1 that possesses a much lower titer (decreased from 10⁹ to 10⁶ pfu/mL) towards the *Microcystis* host". This is a rather cryptic statement and should be reworded to make the meaning clearer.
8. It is very intriguing that Mic1 *TnpB* is apparently important for infection. Does the region of *tnpB* where insertion of *cas9/csn1* gene was found contain sequences with microhomology to the *cas9/csn1* locus?
9. Which method was used to prepare cDNA libraries?

Dear Editor,

Thank you very much for handling our manuscript. We appreciated very much the constructive comments and suggestions of you and the reviewer, which help us to improve the quality of our manuscript. We have addressed all questions and suggestions point by point.

Editor's comments

This manuscript was reviewed and seen and worthy of publication after a few minor modifications. Overall, I found the manuscript well written and presented, although I did find some minor errors in English and grammar such as the following:

Line 43: I would remove "in the long history"

Line 47 replace has ever been with "is"

Line 79 replace been burst with "developed"

If possible, I would suggest the authors have someone proficient in English to check through, but again I would note the errors are small and do not take away from understanding the science of the manuscript.

A: All above points have been revised. According to your suggestion, we have carefully proofread the manuscript and corrected mistakes throughout the text.

Reviewer #1

Freshwater cyanobacteria can cause devastating blooms and their interaction with phages is of considerable scientific and practical interest. In the manuscript, Wang et al. investigated the transcriptome of the cyanobacterium *Microcystis aeruginosa* infected with a previously isolated and sequenced siphovirus Mic1. Based on time-course changes in transcript abundances, phage genes were divided into three classes (early, middle and late) with two subclusters within each group. The authors assigned plausible putative functions for many genes from each temporal classes. They also monitored changes in abundance of host transcripts at different stages of viral infection. A Mic1 phage resistant *M. aeruginosa* strain, as well as an alleviated phage were isolated and mutations responsible were identified.

Below a several comments/questions for the authors.

Q1. The two separate axes as well as separate scales for the host cell number and PFU/ml in Fig. 1A are confusing. Since the logarithmic scale is used, both could be depicted on the same scale.

A: Fig. 1A was revised according to your suggestion.

Q2. Did the authors analyze the structure of viral DNA within the virions, specifically, were the termini of viral DNA determined? In Fig. S2 the genome map starts with the *tnpB* gene. Does it imply that this gene is truly located at the terminus? Given that *tnpB* is an early gene, does this imply that this end is injected first?

A: Although the capsid structure of Mic1 has been solved in 2019 (Jin et al., *Structure*, 2019, DOI: 10.1016/j.str.2019.07.003), the termini of genomic DNA have not yet determined. Supplementary to Fig. 2, the genes in Fig. S2 are sequentially aligned according to the temporal expression patterns identified by RNA-seq. We don't know if *tnpB* is located at the terminus of the viral genomic DNA.

Q3. It is intriguing that Mic1 genome contains quite large apparently non-coding regions. For instance, in the published annotation ~3 kbps region downstream *tnpB* (gene 49) lacks any annotated features. Given that most phage genomes demonstrate very high coding density, it is quite puzzling. Do the authors have any clues on the functional role of this region? Is this region transcriptionally active?

A: Indeed, there is ~3 kbps so-called non-coding region at the downstream of *gp49/tnpB*, which has not been detected in our RNA-seq data. However, we proposed that this region might be transcribed into small RNAs, which are absent in our present sequencing library.

Q4. It is stated that Fig. 3A contains a volcano plot. Usually, a volcano plot shows two values: significance and fold change. In Fig. 3A, only \log_2 fold change axis is shown without the significance. Either the plot should be corrected or the text modified.

A: Thanks for your suggestion. Fig. 3A is not a traditional volcano plot, but an overall display of all differentially expressed genes of the host at different time points. We have revised the legend of Fig. 3A, and added the volcano plot of host differentially

expressed genes at each time point as Fig. S3.

Q5. Replace gene numbers with the names of cas genes (e. g. csm6 instead 1109) in Fig. 3b.

A: Revised.

Q6. In the Materials and Methods section, it is stated that differentially expressed host genes were identified using DESeq2 package. Results of the analysis, including adjusted p-values, should be provided. Currently, only log₂FC values (e. g., Table S3) are shown.

A: Added as the Table S7, of which the log₂FC values, *p*-values and adjusted *p*-values of all differentially expressed host genes were provided.

Q7. Line 262 states "we also isolated a strain of Mic1 that possesses a much lower titer (decreased from 10⁹ to 10⁶ pfu/mL) towards the *Microcystis* host". This is a rather cryptic statement and should be reworded to make the meaning clearer.

A: Sorry for the unclear description. We have revised this sentence.

Q8. It is very intriguing that Mic1 TnpB is apparently important for infection. Does the region of tnpB where insertion of cas9/csn1 gene was found contain sequences with microhomology to the cas9/csn1 locus?

A: Sorry for the mistake. According HHpred, the ORF2067 of *Microcystis aeruginosa*

FACHB 1339 was previously mis-annotated as a type II CRISPR endonuclease Cas9/Csn1. During revision, via structure prediction and comparison, we found that it shares a similar structure to the nickase IsrB (Fig. S6), with an RMSD of 2.3 Å over 315 Cα atoms. Previous studies revealed that IsrB, a member of the IS200/IS605 superfamily of transposons, is usually responsible for nicking the non-target strand of dsDNA containing a 5'-NTGA-3' target-adjacent motif (Altae-Tran et al., *Science*, 2021, DOI: 10.1126/science.abj6856). Indeed, *orf2067* is inserted at the motif 5'-ATGA-3', corresponding to 263 to 266 of original *tnpB*.

Q9. Which method was used to prepare cDNA libraries?

A: Added in the “Materials and Methods” section.

Re: Spectrum00298-24R1 (Profiling the interplay and coevolution of *Microcystis aeruginosa* and cyanosiphophage Mic1)

Dear Prof. Cong-Zhao Zhou:

Thank you for your revisions.

Your manuscript has been accepted, and I am forwarding it to the ASM production staff for publication. Your paper will first be checked to make sure all elements meet the technical requirements. ASM staff will contact you if anything needs to be revised before copyediting and production can begin. Otherwise, you will be notified when your proofs are ready to be viewed.

Sincerely,
Blair Steven
Editor
Microbiology Spectrum